# Physics-Informed Neural Networks with Learnable Loss Balancing and Transfer Learning

## Abstract

We propose a self-supervised physics-informed neural network (PINN) framework that adaptively balances physics-based and data-driven supervision for scientific machine learning under data scarcity. Unlike prior PINNs that rely on fixed or heuristic weighting of physics residuals and data loss, our approach introduces a learnable blending neuron that dynamically adjusts the relative contribution of each term based on their uncertainties. This mechanism enables stable training and improved generalization without manual tuning. To further enhance efficiency, we integrate a transfer learning strategy that reuses representations from related domains and adapts them to new physical systems with limited data. We validate the framework for the prediction of heat transfer in liquid-metal miniature heat sinks using only 87 CFD datapoints, where the adaptive PINN achieves an error $< 8\%$, outperforming shallow neural networks, kernel methods, and physics-only baselines. Our framework provides a general recipe for embedding physics adaptively into neural networks, offering a robust and reproducible approach for data-scarce problems across various scientific domains, including fluid dynamics and material modeling.

## 1 Introduction

Scientific machine learning is increasingly solving problems where only limited data is available, such as turbulence modeling, climate forecasting, and thermal transport in novel energy systems. Traditional data-driven neural networks excel when large datasets are present, but their accuracy and stability degrade when observations are scarce or noisy. In such cases, the incorporation of governing physics into the learning pipeline has shown a promising approach Sharma et al. (2023); Guastoni et al. (2021). Physics-informed neural networks (PINNs) and related frameworks integrate physical residuals directly into the loss function, constraining models to respect conservation laws and differential equations.

A key challenge in PINNs is balancing the contributions of data-driven and physics-based losses. Fixed or manually tuned weights can lead to poor convergence or biased predictions. Prior studies have investigated adaptive strategies, including self-adaptive PINNs McClenny & Braga-Neto (2020), gradient-normalization methods Wang et al. (2021), and uncertainty-driven weighting in Bayesian PINNs Yang et al. (2021). Despite these advances, most approaches require either heuristic rules, Bayesian overhead, or remain sensitive to hyperparameter choices. This motivates the need for a lightweight, self-supervised mechanism that dynamically adjusts the physics–data trade-off during training.

In parallel, transfer learning has gained prominence in both general ML Zhuang et al. (2020); Liu et al. (2019) and scientific domains Jeon et al. (2022a;b), enabling knowledge reuse across related tasks. For thermal-fluid applications, transfer learning has been used to accelerate CFD surrogates Baghban et al. (2019); Pourghasemi & Fathi (2023), but its integration with PINNs remains underexplored. Combining adaptive physics–data weighting with transfer learning offers the potential for robust learning even in data-scarce regimes.

In this work, we propose a self-supervised PINN framework with a *learnable blending neuron* that dynamically balances physics residuals and data losses based on their uncertainties. We further

incorporate a transfer learning scheme that reuses hidden-layer representations from related domains to accelerate training in new physical systems. To evaluate the method, we consider the prediction of convective heat transfer in sodium-cooled miniature heat sinks using only 87 CFD datapoints, a regime where high-fidelity simulation is prohibitively costly. Our contributions are threefold:

- We introduce a simple yet effective self-supervised mechanism for adaptive loss balancing in PINNs, removing the need for manual tuning or Bayesian complexity.

- We use transfer learning, demonstrating its effectiveness in scientific ML tasks with scarce data.

- We validate the framework on a challenging liquid-metal heat transfer problem and benchmark against shallow neural networks, kernel methods, and physics-only baselines.

This framework provides a general recipe for embedding physics adaptively into neural networks, with implications for a wide range of domains including heat transfer, materials science, and aerospace engineering.

## 2 RELATED WORK

**Machine learning for thermal–fluid systems.** Data-driven models have been widely applied in fluid mechanics and thermal sciences, particularly for predicting heat transfer coefficients and fluid properties when experimental or CFD data are limited. Examples include neural-network and kernel-based surrogates for nanofluid flows Baghban et al. (2019); Tafarroj et al. (2017); Kurt & Kayfeci (2009); Yousefi et al. (2012), convective heat transfer prediction in coils and microchannels Baghban et al. (2016); Bhattacharya et al. (2022), and convolutional-network approaches for wall-bounded turbulence Guastoni et al. (2021). More recent work has combined ML with high-fidelity CFD solvers to accelerate simulations Jeon et al. (2022b); Pourghasemi & Fathi (2023). Comprehensive reviews of physics-informed ML in fluid mechanics emphasize the potential of integrating physics into learning pipelines Sharma et al. (2023).

**Physics-informed neural networks.** PINNs incorporate governing equations as soft constraints, improving generalization under data scarcity. However, balancing the relative contributions of physics and data remains challenging. Several adaptive schemes have been proposed: self-adaptive PINNs using gradient information McClenny & Braga-Neto (2020), NTK-based analyses of PINN training pathologies Wang et al. (2021), and Bayesian approaches that weight losses according to uncertainty Yang et al. (2021). These methods improve training stability but often require heuristic tuning, additional complexity, or Bayesian overhead. Our approach differs by introducing a simple *learnable blending neuron* that dynamically adjusts physics–data weighting in a fully self-supervised manner.

**Transfer learning in scientific ML.** Transfer learning has achieved success across domains from computer vision to geoscience Zhuang et al. (2020); Liu et al. (2019). In thermal–fluid contexts, transfer strategies have been applied to accelerate unsteady CFD simulations Jeon et al. (2022a) and other flow-physics surrogates. Nevertheless, integration of transfer learning into PINNs remains underexplored. Our framework bridges this gap by combining transfer learning with adaptive physics–data balancing, enabling PINNs to efficiently adapt knowledge across related physical systems.

**Positioning of this work.** In summary, while prior work has investigated adaptive weighting in PINNs McClenny & Braga-Neto (2020); Wang et al. (2021); Yang et al. (2021) and physics-informed transfer learning for fluid simulations Jeon et al. (2022a), our contribution unifies these directions. We present a *self-supervised adaptive PINN with transfer learning*, validated on a challenging small-data case of sodium-cooled miniature heat sinks. This combination yields a robust and lightweight framework for scientific ML under data scarcity, complementing and extending prior approaches.

## 3 METHODOLOGY

### 3.1 CFD SIMULATIONS

Computational Fluid Dynamics (CFD) simulations were conducted to generate a dataset of 87 data points. These simulations follow the numerical framework described in Pourghasemi & Fathi (2023) and were executed using ANSYS FLUENT. The objective was to obtain precise Nusselt numbers for both laminar and turbulent flow of liquid sodium in stainless steel (SS-316) rectangular miniature heat sinks under varying physical conditions.

The input parameters spanned a wide range, including heat sink width, aspect ratio, hydraulic diameter, and Peclet number of the sodium coolant. The governing equations included the fundamental equations of incompressible, steady-state flow: the continuity equation, the Navier–Stokes momentum equations, and the energy conservation equation. Additionally, heat conduction within the solid substrate was modeled with a temperature-dependent conductivity. These equations are summarized as follows:

**Continuity equation (incompressible flow):**
$$\nabla \cdot (\rho \mathbf{u}) = 0, \tag{1}$$
where $\rho$ is the fluid density and $\mathbf{u}$ is the velocity vector.

**Navier–Stokes momentum equation:**
$$\nabla \cdot (\rho \mathbf{u}\mathbf{u}) = -\nabla P + \nabla \cdot \left(\mu(\nabla \mathbf{u} + \nabla^T \mathbf{u})\right) + \rho \mathbf{g}, \tag{2}$$
where $P$ is pressure, $\mu$ is dynamic viscosity, and $\mathbf{g}$ is the gravitational acceleration.

**Energy equation (fluid):**
$$\nabla \cdot (\rho c_p \mathbf{u} T) = \nabla \cdot (k_f \nabla T), \tag{3}$$
where $c_p$ is the specific heat capacity, $T$ is the temperature field, and $k_f$ is the thermal conductivity of the fluid.

**Heat conduction equation (solid substrate):**
$$\nabla \cdot (k_s \nabla T) = 0, \tag{4}$$
where $k_s$ is the thermal conductivity of the solid material.

The study employed steady-state numerical simulations with a no-slip boundary condition at the solid–fluid interfaces of the miniature heat sinks. The coolant was introduced at uniform velocity and constant inlet temperature.

### 3.2 SELF-SUPERVISED ADAPTIVE PINN FRAMEWORK

The machine learning framework is based on physics-informed neural networks (PINNs), where the governing partial differential equations (PDEs) of fluid flow and heat transfer are embedded as soft constraints in the training objective. Let $\theta$ denote the network parameters. The standard PINN loss is a weighted sum of data-driven and physics-driven terms:
$$\mathcal{L}(\theta) = \lambda_d \, \mathcal{L}_{\text{data}}(\theta) + \lambda_p \, \mathcal{L}_{\text{physics}}(\theta), \tag{5}$$
where $\mathcal{L}_{\text{data}}$ represents the discrepancy between network predictions and available CFD data, and $\mathcal{L}_{\text{physics}}$ measures PDE residuals from Equations 1–4. The coefficients $\lambda_d$ and $\lambda_p$ balance the contributions of the two terms.

**Adaptive blending neuron.** Instead of fixing $\lambda_d$ and $\lambda_p$ manually, we introduce a learnable *blending neuron* that adaptively adjusts their relative contributions during training. Specifically,
$$\lambda_d = \sigma(\alpha), \qquad \lambda_p = 1 - \sigma(\alpha), \tag{6}$$
where $\sigma(\cdot)$ is the sigmoid function and $\alpha$ is a trainable scalar parameter. This formulation ensures $0 < \lambda_d, \lambda_p < 1$ and allows the model to automatically discover the optimal balance between physics and data supervision. During training, $\alpha$ is updated by backpropagation along with $\theta$, making the weighting self-supervised.

Table 1: Performance comparison of GP and SVR models using RBF kernel.

| Method | Kernel | MAPE |
|---|---|---|
| GP | RBF | 0.0756 |
| SVR–RS | RBF | 0.0272 |
| SVR–Bayesian | RBF | 0.0125 |

**Data-driven loss.** The data loss is defined as the mean squared error (MSE) between predicted and CFD-computed Nusselt numbers:

$$\mathcal{L}_{\text{data}}(\theta) = \frac{1}{N_d} \sum_{i=1}^{N_d} \left( \hat{y}_i(\theta) - y_i \right)^2,$$ 
(7)

where $y_i$ are ground-truth CFD values and $\hat{y}_i(\theta)$ are PINN predictions at the same inputs.

**Physics residual loss.** The physics loss is constructed from PDE residuals evaluated at collocation points $\{\mathbf{x}_j\}_{j=1}^{N_p}$:

$$\mathcal{L}_{\text{physics}}(\theta) = \frac{1}{N_p} \sum_{j=1}^{N_p} \left( \mathcal{R}(\mathbf{x}_j; \theta) \right)^2,$$ 
(8)

where $\mathcal{R}$ denotes the residual of the governing equations (Equations 1–4) computed with PINN-predicted velocity, pressure, and temperature fields.

### 3.3 TRANSFER LEARNING FOR DATA-SCARCE REGIMES

To further enhance learning efficiency, we incorporate a transfer learning (TL) strategy. A base PINN is first trained on a source dataset (e.g., water-cooled microchannels), where larger training data are available. The network parameters $\theta^*$ from the source task are then used to initialize the target PINN for sodium-cooled miniature heat sinks. Specifically,

$$\theta_{\text{target}}^{(0)} \leftarrow \theta_{\text{source}}^*.$$ 
(9)

During fine-tuning, only the last few layers and the blending neuron parameter $\alpha$ are updated, while early layers retain transferable low-level representations. This approach reduces training time and improves convergence stability under extremely small target datasets (87 CFD points).

### 3.4 TRAINING PROCEDURE

The overall training process alternates between minimizing $\mathcal{L}_{\text{data}}$ and $\mathcal{L}_{\text{physics}}$, with weights governed by the adaptive blending neuron. A schematic of the framework is shown in Figure 1. The Adam optimizer was employed with learning rate scheduling, and early stopping was used to prevent overfitting. Monte Carlo cross-validation Shan (2022); Elmessiry et al. (2017) was applied to quantify generalization performance and statistical robustness.

## 4 RESULTS

### 4.1 GAUSSIAN PROCESS AND SUPPORT VECTOR REGRESSION

Table 1 compares Gaussian Process (GP) and Support Vector Regression (SVR) models using a radial basis function (RBF) kernel. SVR consistently outperforms GP due to its ability to exploit data distribution more effectively and tune additional hyperparameters for extrapolation. Among SVR models, randomized search (RS) optimization achieves a minimum mean absolute percentage error (MAPE) of 2.72% after four iterations, while Bayesian optimization converges faster, reaching a lower MAPE of 1.25% within 17 iterations.

The corresponding SVR hyperparameters are reported in Table 2, where Bayesian optimization selects a higher regularization parameter ($C = 27.23$) compared to RS ($C = 10$), further contributing to its superior generalization.

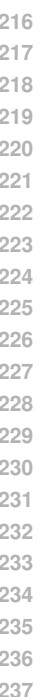
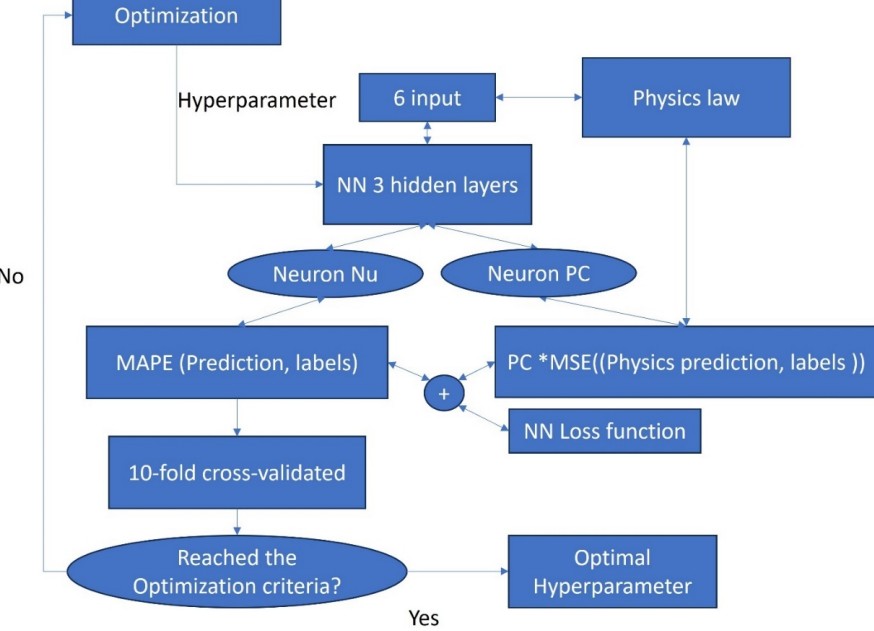

Figure 1: Schematic of the proposed self-supervised adaptive PINN with transfer learning. The blending neuron learns to weight physics residuals and data loss.

Table 2: Optimal hyperparameters for SVR under RS and Bayesian optimization.

| Method | Kernel | $C$ | $\gamma$ | $\epsilon$ | MAPE |
|---|---|---|---|---|---|
| SVR–RS | RBF | 10 | 0.0001 | 0.0010 | 0.0272 |
| SVR–Bayesian | RBF | 27.23 | 0.0007 | 0.0031 | 0.0125 |

### 4.2 TRANSFER LEARNING FOR NEURAL NETWORKS

Transfer learning (TL) further improves neural network performance. Table 3 shows that transferring the first hidden layer from a water-trained network reduces MAPE from 0.0028 (no transfer) to 0.0020. Genetic Algorithm (GA)-based optimization selected an optimal architecture of two hidden layers, with three neurons transferred from the source network and eight randomly initialized neurons in the second layer. Figure 2 confirms that transferring earlier layers captures generalizable features, while transferring layers closer to the output degrades accuracy due to domain-specific representations.

The statistical difference between water and sodium datasets is illustrated in Figure 3.

Kernel Density Estimates (Figure 3) confirm a broader variance for sodium data. A Mann–Whitney U test rejects the null hypothesis ($p \ll 0.05$), supporting the suitability of TL from water to sodium.

### 4.3 SELF-SUPERVISED PINN PERFORMANCE

Bayesian optimization determined the optimal architecture of the self-supervised PINN to be two hidden layers of 20 neurons and one hidden layer of 12 neurons, with Adam optimizer at learning rate 0.34. The 10-fold cross-validated MAPE was 0.0185 after 100 optimization iterations. Figure 4 shows the distribution of the physics coefficient neuron $\lambda_p$, centered around 0.5, confirming balanced contributions of physics and data.

Table 3: Impact of transfer learning on neural network performance.

| Method | MAPE |
|---|---|
| No Transfer | 0.0028 |
| Transfer (1st hidden layer) | **0.0020** |

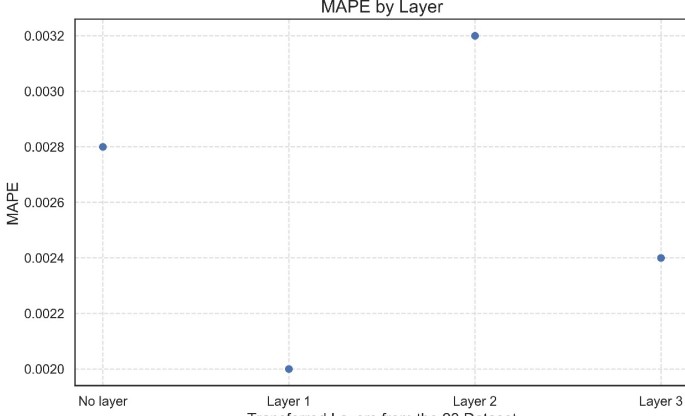

Figure 2: Effect of transfer learning from different layers. The first-layer transfer achieves the lowest error, while transferring deeper layers closer to the output increases error.

## 4.4 BENCHMARKING AND VALIDATION

Table 4 reports benchmarking results across all methods. SVR–Bayesian achieves the lowest error among classical ML methods (MAPE = 0.0125), while the adaptive NN with transfer learning achieves the overall lowest error (MAPE = 0.0020). The self-supervised PINN achieves competitive performance (MAPE = 0.0185) but demonstrates superior robustness.

Monte Carlo simulations (500 trials) were further used to analyze robustness (Table 5). While the PINN exhibited higher variance in MAPE during training, its prediction variance on the holdout dataset was lower than both baseline NN (no transfer) and kernel methods. This indicates improved robustness to hyperparameter and initialization randomness.

Figures 5–6 visualize predictions on holdout sets. Kernel-based methods achieve reasonable accuracy but tend to underfit, whereas NN-based methods capture more variance. The self-supervised PINN remains consistently within the $\pm 8\%$ error margin, validating its robustness for Nusselt number prediction in sodium heat sinks.

## 5 DISCUSSION

The results demonstrate that classical kernel-based methods such as GP and SVR provide competitive baselines, with SVR–Bayesian achieving a MAPE of $0.0125$. However, their performance is limited by sensitivity to kernel choice and hyperparameter tuning. In contrast, neural-network approaches, particularly those incorporating transfer learning, achieve substantially lower errors $(0.0020)$ by reusing generalizable representations from water-cooled datasets. This highlights the potential of cross-domain transfer in scientific ML, where related physical systems often share underlying structural features.

The self-supervised PINN achieves a higher MAPE $(0.0185)$ compared to the best NN and SVR models, yet exhibits superior robustness during Monte Carlo validation. Specifically, it maintains lower prediction variance on holdout datasets despite larger variance during training. This robustness arises from the adaptive blending neuron, which balances physics-based and data-driven supervision

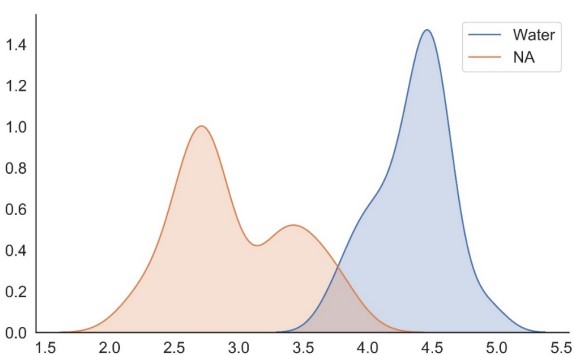

Figure 3: KDE comparison of water and sodium Nusselt numbers. Sodium exhibits higher variance than water, supporting transfer learning across domains.

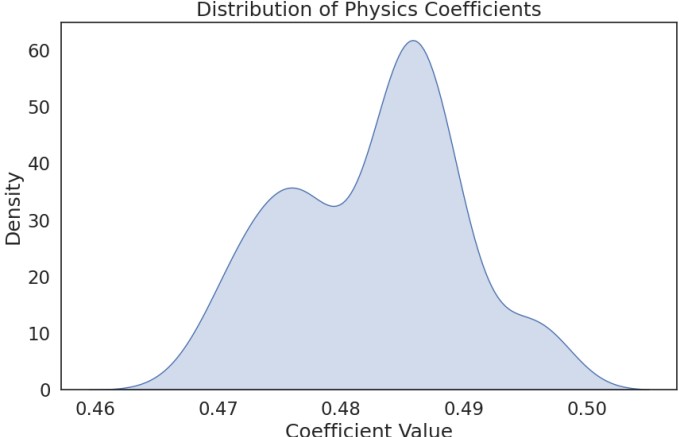

Figure 4: Distribution of the learned physics coefficient neuron in the self-supervised PINN, centered near 0.5.

without requiring manual tuning. The learned coefficient distribution (centered around 0.5) confirms that the network adaptively exploits both sources of information.

An important insight is that the adaptive PINN, although not the lowest in raw error, provides a more reliable and interpretable framework for deployment in data-scarce regimes. When CFD data availability is limited, the physics component stabilizes learning, reducing overfitting and improving generalization. The trade-off between raw accuracy and robustness is particularly relevant in safety-critical applications such as thermal management of liquid-metal systems, aerospace vehicles, and biomedical devices.

## 6 CONCLUSION

We introduced a self-supervised PINN framework that adaptively balances physics residuals and data-driven errors through a learnable blending neuron, and we combined this with transfer learning to enhance performance under extreme data scarcity. Our main findings are:

- SVR with Bayesian optimization provides strong baselines but requires extensive hyperparameter tuning.
- Neural networks benefit significantly from transfer learning, with first-layer transfer achieving the lowest MAPE (0.0020).

Table 4: MAPE comparison across ML methods for Nusselt number prediction.

| Model | MAPE |
|---|---|
| NN with Transfer Learning | **0.0020** |
| NN (no transfer) | 0.0028 |
| Self-supervised PINN | 0.0185 |
| Gaussian Process (GP) | 0.0756 |
| SVR–RS | 0.0272 |
| SVR–Bayesian | 0.0125 |

Table 5: Monte Carlo validation metrics across ML methods. TR_NN = transfer learning NN.

| Model | Max var(pred) | Max var(MAPE) | Avg. Epochs |
|---|---|---|---|
| TR_NN | 0.2183 | 0.0002 | 14.30 |
| NN (no transfer) | 0.2323 | 0.0002 | 14.20 |
| Self-supervised PINN | 0.1146 | 0.0008 | 35.00 |

- The self-supervised PINN achieves competitive accuracy (0.0185 MAPE) while demonstrating superior robustness to hyperparameter and initialization randomness.
- Statistical analysis of Nusselt number distributions confirms the validity of transferring representations from water to sodium domains.

Overall, our framework provides a robust approach toward small-data scientific ML, offering both accuracy and reliability. While demonstrated on sodium-cooled miniature heat sinks, the approach generalizes to other domains where governing equations are available but data are limited. Future work includes extending the blending mechanism to multi-physics scenarios, incorporating uncertainty quantification, and scaling the framework to large-scale 3D CFD simulations.

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

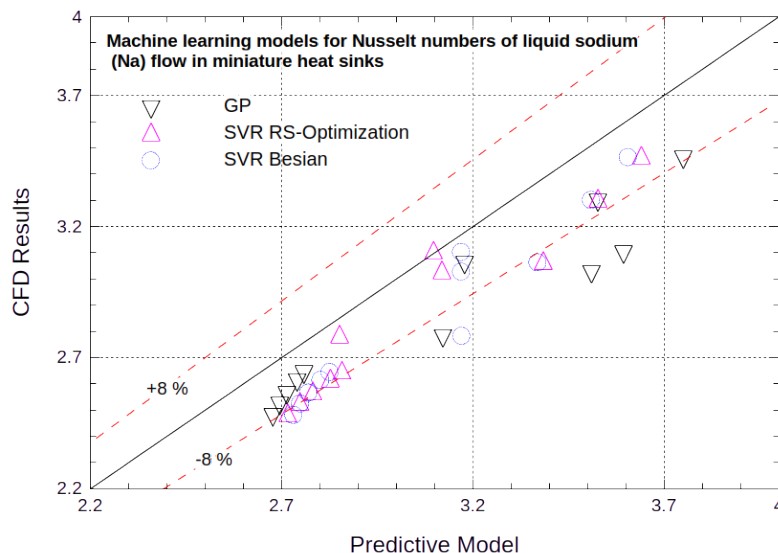

Figure 5: GP and SVR predictions on holdout dataset. SVR–Bayesian achieves better fit than GP, though both remain within $\pm 8\%$ error margin.

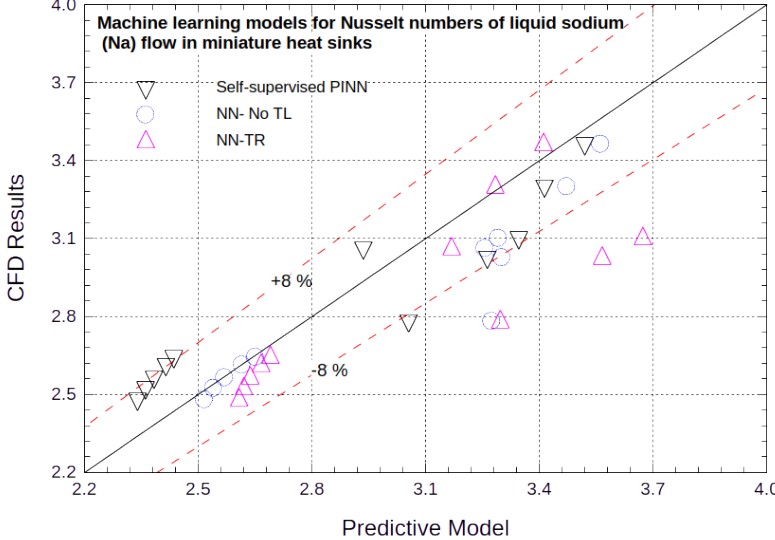

Figure 6: Neural network predictions on holdout dataset. The self-supervised PINN provides the most robust estimations within $\pm 8\%$ margin.

J. Liu, K. Chen, G. Xu, X. Sun, M. Yan, W. Diao, and H. Han. Convolutional neural network-based transfer learning for optical aerial images change detection. *IEEE Geoscience and Remote Sensing Letters*, 17(1):127–131, 2019.

Levi McClenny and Ulisses Braga-Neto. Self-adaptive physics-informed neural networks, 2020.

M. Pourghasemi and N. Fathi. Enhancement of liquid sodium (na) forced convection within miniature heat sinks. *ASME Journal of Heat and Mass Transfer*, 145(5), 2023.

G. Shan. Monte carlo cross-validation for a study with binary outcome and limited sample size. *BMC Medical Informatics and Decision Making*, 22(1):1–15, 2022.

Pushan Sharma, Wai Tong Chung, Bassem Akoush, and Matthias Ihme. A review of physics-informed machine learning in fluid mechanics. *Energies*, 16(5):2343, 2023.

M.M. Tafarroj, O. Mahian, A. Kasaeian, K. Sakamatapan, A.S. Dalkilic, and S. Wongwises. Artificial neural network modeling of nanofluid flow in a microchannel heat sink using experimental data. *International Communications in Heat and Mass Transfer*, 86:25–31, 2017.

Sifan Wang, Yujun Teng, and Paris Perdikaris. When and why pinns fail to train: A neural tangent kernel perspective. In *International Conference on Learning Representations (ICLR)*, 2021.

Liu Yang, Xuhui Meng, and George Em Karniadakis. B-pinns: Bayesian physics-informed neural networks for forward and inverse pde problems with noisy data, 2021.

F. Yousefi, H. Karimi, and M.M. Papari. Modeling viscosity of nanofluids using diffusional neural networks. *Journal of Molecular Liquids*, 175:85–90, 2012.

F. Zhuang, Z. Qi, K. Duan, D. Xi, Y. Zhu, H. Zhu, H. Xiong, and Q. He. A comprehensive survey on transfer learning. *Proceedings of the IEEE*, 109(1):43–76, 2020.

