# OpenReview forum: "Physics-Informed Neural Networks with Learnable Loss Balancing and Transfer Learning"
_ICLR.cc/2026/Conference — Submitted to ICLR 2026_

### Official Review · Reviewer_knDQ · 2025-10-15

**Soundness:** 1
**Presentation:** 1
**Contribution:** 1
**Rating:** 0
**Confidence:** 5

**Summary:**

This paper proposes a self-supervised physics-informed neural network (PINN) that adaptively balances data-driven and physics-based losses through a learnable “blending neuron,” removing the need for manual weight tuning between the two. The authors further combine this mechanism with transfer learning to improve generalization under extreme data scarcity. The paper reports improved robustness and comparable accuracy to kernel and standard NN baselines, claiming that the adaptive PINN provides a reliable, interpretable approach for small-data scientific ML tasks.

**Strengths:**

In my opinion, this work is not ready to publish. (No Strengths)

**Weaknesses:**

**Weakness**

- The manuscript suffers from serious readability issues and lacks proper organization. Tables and figures are not placed where they are referenced in the text, and some even appear in inappropriate sections. For instance, Figures 5 and 6 are inserted between the reference entries.


- The references are poorly managed. The paper fails to cite the seminal work that originally proposed Physics-Informed Neural Networks [1]. Moreover, one of the cited works, “Sifan Wang, Yujun Teng, and Paris Perdikaris. *When and why PINNs fail to train: A neural tangent kernel perspective.* In International Conference on Learning Representations (ICLR), 2021.”, is incorrectly labeled as an ICLR publication, when in fact it was published in the *Journal of Computational Physics* in 2022 [2].

- The experimental section is severely underdeveloped. The number of experiments is too small to support the claims, and key implementation details such as the experimental settings, boundary conditions, sampling procedures, and model configurations are inadequately described.

[1] Raissi, Maziar, Paris Perdikaris, and George E. Karniadakis. "Physics-informed neural networks: A deep learning framework for solving forward and inverse problems involving nonlinear partial differential equations." Journal of Computational physics 378 (2019): 686-707.

[2] Wang, Sifan, Xinling Yu, and Paris Perdikaris. "When and why PINNs fail to train: A neural tangent kernel perspective." Journal of Computational Physics 449 (2022): 110768.

**Questions:**

No questions.

---

### Official Review · Reviewer_TGbo · 2025-10-18

**Soundness:** 2
**Presentation:** 1
**Contribution:** 1
**Rating:** 2
**Confidence:** 5

**Summary:**

This paper proposes a self-supervised physics-informed neural network (PINN) framework that adaptively balances physics-based and data-driven supervision for scientific machine learning under data scarcity.

**Strengths:**

The paper needs significant revisions.

**Weaknesses:**

1. The experiments do not adequately support the paper's claims. For instance, the paper asserts that the proposed approach enables stable training and improved generalization without manual tuning, yet I found no experimental results demonstrating these capabilities.

2. The paper's organization and writing quality are inadequate. More critically, given that adaptive weights are presented as a key novelty, the paper should formally describe this mechanism as its main technical content. However, I could not find any rigorous mathematical formulation of this approach anywhere in the manuscript.

3. In its current form, this work reads more like an incomplete undergraduate student's senior project than a publication-ready contribution. The paper requires substantial revision before it can be considered suitable for publication.

**Questions:**

No questions.

---

### Official Review · Reviewer_BFCN · 2025-10-27

**Soundness:** 2
**Presentation:** 1
**Contribution:** 1
**Rating:** 2
**Confidence:** 4

**Summary:**

The paper introduces several techniques to improve training of PINNs for CFD problems. The paper introduces blending neuron which are used to balance between each type of training loss in PINN training, and perform transfer learning to initialise the PINN parameters which that trained in another similar pINN problem. The paper shows how these techniques can improve training of PINNs on CFD problems.

**Strengths:**

The problem considered is real, since PINNs can be difficult to train especially on CFD cases.

There seems to be some results that demonstrate the use of the proposed methods are able to improve on when the methods are not used.

**Weaknesses:**

The paper only considers CFD problems, which is a much smaller scope than what the paper title suggests. It would be much more convincing to also perform the same loss balancing mechanism on other PINN training scenarios as well.

Additionally, it is not clear how the methods in the paper are novel. Balancing of loss functions are already done in more systematic ways in existing PINN works, and transfer learning for PINNs are already a setting that have been considered in other PINN works. A better description to how this paper is beyond just a blend of existing methods would be appreciated.

Sec 3.1 -- depending on the scope of the problem this part may not be necessary in the main text and better relegated to the appendix instead. Some parts within Sec 3.2 are also the same, since are common in PINN training anyway.

Fig 1 -- flow chart is confusing to follow. Would prefer it written as a pseudocode, or at least to have the accompanying text (Sec 3.4) to describe the algorithm more clearly.

All results lack error bars.

The experiments are insufficient. There are other methods for training to improve PINN convergence which involves balancing of loss functions which are not considered by the paper. Comparisons against these benchmarks would make the paper more convincing.

The reported results are poorly explained. The MAPE are given, but the error of which quantity is being reported here? This part is not mentioned in the paper. Furthermore, in transfer learning settings, what is the transfer learning being done from, and to which setting?

Figs 5 and 6 -- not clear as to what the different points mean. Results may be better shown as a table.

**Questions:**

1. Around Line 156 -- the use of neurons are not clear as to why they are used. If they are just back-propped, would the NN just not prioritise the loss factor that are smaller, hence ignoring the large error term and making that part not learned? This aspect is unclear, and clarifications should have been added to the paper.

2. Related to the above -- what values of the neurons are learned? How does alpha evolve throughout the training process? Can any interpretation be taken from this value, and how would it compare with the papers that manually tune this or automatically select based on other heuristics?

3. How is the training time when the blending neurons are used, compared to using existing PINN training methods? Additional results on resource usage would be interesting.

4. What are the interpretations for Fig 3 and 4? Are these KDE values used during the training anywhere or are just for better understanding for the training process?

---

### Official Review · Reviewer_ggPw · 2025-10-31

**Soundness:** 2
**Presentation:** 3
**Contribution:** 2
**Rating:** 2
**Confidence:** 3

**Summary:**

This paper proposes a self-supervised PINN with a learnable neuron that adaptively balances data and physics losses, as well as uses transfer learning techniques to reuse representations from related physical domains and fine tune only layers specific to new physical systems. While the implementation is straightforward, the novelty of the contribution is limited.

**Strengths:**

1. The workflow and method section is easy to understand.
2. Clearly presented the motivation and experimental settings.

**Weaknesses:**

1. The experimental section should also compare against other self-weighting methods as baselines. In addition, several important PINN variants are missing from the baselines, such as hard-constrained approaches.

2. The experimental section includes only one example. While valuable, it is narrow in scope.

3. PINNs can fail when the PDE coefficient is large, leading to ill-conditioning. Additional experiments demonstrating how the proposed method mitigates this issue would be better.

**Questions:**

I am not sure why this paper compares against GP and SVR. As I understand it, these are not neural methods for solving forward problems.

**Details Of Ethics Concerns:**

None.

---

### Meta-Review · Area_Chair_tChw · 2026-01-07

**Summary:**

All reviewers recommend rejection, with consistent concerns regarding novelty, experimental depth, and presentation quality. While the application is relevant, the technical contribution is limited. The proposed adaptive loss balancing reduces to learning a single scalar parameter and is not convincingly distinguished from existing self-adaptive or uncertainty-based PINN weighting methods. No theoretical analysis or strong empirical evidence is provided to justify the claimed advantages.

The experimental evaluation is narrow and insufficient to support the paper’s claims. Results are limited to a single CFD problem, and important PINN baselines are missing. Comparisons focus heavily on GP and SVR, which are not standard for PINNs. Claims of improved robustness and stability are not properly demonstrated.

The transfer learning component is also underdeveloped, with limited analysis of when and why transfer is effective, and a lack of connection to prior PINN transfer learning work. Finally, the paper suffers from serious clarity and organization issues, including confusing figures, missing details, and citation problems.

Overall, despite a reasonable motivation, the submission lacks a clear, novel contribution and convincing experimental support, and does not meet the standards for ICLR. I therefore recommend rejection.

**Reviewer Concerns:**

No concerns were addressed.

**Reviewer Scores:**

No change in scores.

---

### Decision · Program_Chairs · 2026-01-26

Reject